# DDX43 mRNA expression and protein levels in relation to clinicopathological profile of breast cancer

**Noha N. Amer**[1]*, **Rabab Khairat**[2©], **Amal M. Hammad**[3©], **Mahmoud M. Kamel**[4,5©]*

**1** Faculty of Pharmacy (Girls), Department of Biochemistry and Molecular Biology, Al-Azhar University, Cairo, Egypt, **2** Medical Molecular Genetics Department, Human Genetics and Genomic Research Division, National Research Center, Cairo, Egypt, **3** Faculty of medicine, Department of Medical Biochemistry, Al-Azhar University, Damietta, Egypt, **4** Clinical Pathology Department, National Cancer Institute, Cairo, Egypt, **5** Baheya Centre for Early Detection and Treatment of Breast Cancer, Giza, Egypt

© These authors contributed equally to this work.
* noha_amer@azhar.edu.eg (NNA); mahmoud.kamel@nci.cu.edu.eg (MMK)

## Abstract

### Background

Breast cancer (BC) is the most often diagnosed cancer in women globally. Cancer cells appear to rely heavily on RNA helicases. DDX43 is one of DEAD- box RNA helicase family members. But, the relationship between clinicopathological, prognostic significance in different BC subtypes and DDX43 expression remains unclear. Therefore, the purpose of this study was to assess the clinicopathological significance of DDX43 protein and mRNA expression in different BC subtypes.

### Materials and methods

A total of 80 females newly diagnosed with BC and 20 control females that were age-matched were recruited for this study. DDX43 protein levels were measured by ELISA technique. We used a real-time polymerase chain reaction quantification (real-time PCR) to measure the levels of DDX43 mRNA expression. Levels of DDX43 protein and mRNA expression within BC patients had been compared to those of control subjects and correlated with clinicopathological data.

### Results

The mean normalized serum levels of DDX43 protein were slightly higher in control than in both benign and malignant groups, but this result was non-significant. The mean normalized level of DDX43 mRNA expression was higher in the control than in both benign and malignant cases, although the results were not statistically significant and marginally significant, respectively. Moreover, the mean normalized level of DDX43 mRNA expression was significantly higher in benign than in malignant cases. In malignant cases, low DDX43 protein expression was linked to higher nuclear grade and invasive duct carcinoma (IDC), whereas

**Funding:** The author(s) received no specific funding for this work.

**Competing interests:** The authors have declared that no competing interests exist.

high mRNA expression was linked to the aggressive types of breast cancer such as TNBC, higher tumor and nuclear grades.

## Conclusion

This study explored the potential of using blood DDX43 mRNA expression or protein levels, or both in clinical settings as a marker of disease progression in human breast cancer. DDX43 mRNA expression proposes a less invasive method for discriminating benign from malignant BC.

## Introduction

The epithelial cells that line the breast's ducts and lobules proliferate malignantly in breast cancer (BC) [1]. Breast cancer is the fifth biggest cause of cancer death worldwide. The most prevalent type of cancer in women is breast cancer, affecting one out of every four cases and killing one out of every six deaths. In the vast majority of countries, it is the most prevalent cause of death and first cancer in terms of incidence [2]. According to GLOBOCAN data, new cases and deaths of breast cancer constitute 32.4% and 22.4% of female cancer cases in 2020 in Egypt [3].

Breast cancer is clinically classified into four subgroups due to its high degree of variability: Luminal A, Luminal B, human epidermal growth factor receptor-2 (HER2)-positive, and triple-negative (TNBC) [4]. Incidence, prognosis, diagnosis age, and treatment response vary widely within these intrinsic subgroups [5].

The molecular pathways causing breast cancer are still unknown despite the vast molecular studies that have been conducted on its diagnosis and therapy [6,7]. Therefore, it should be a top focus to find possible treatment targets and biomarkers for prognosis and diagnosis [8]. But the evolution of tumor molecular biology has also encouraged the development of novel prediction techniques on the basis of genes associated with prognosis. A more precise estimate of an individual's survival may be made by using some prognostic markers that reflect tumor development at the molecular level.

Remarkably, cancer cells appear to depend intensely on RNA helicases to satisfy the augmented overall protein production request as well as to promote survival by translating some pro-oncogenic mRNAs. RNA helicases play other parts in cancer biology by regulating transcription and alternative splicing (for example DDX5, and DDX17), ribosomal biogenesis (for example DDX5, DDX21, and DDX43), mRNA transport (for example DDX5, DDX3), miRNA regulation (for example DDX3, DDX5) as well as apoptosis (for example DDX3, RHA), and other processes [9].

In a human sarcoma cell line, researchers found the cancer/testis antigen gene DDX43, commonly known as HAGE (helicase antigen gene). A 73-kDa protein from ATP-dependent RNA helicase DEAD-box family members is encoded by this gene, which is located on chromosome 6 (6q12-q13). In tumors, mRNA levels of DDX43 are not less than 100 times greater than in normal tissues. Numerous types of cancerous cells contain varying amounts of DDX43 such as the brain, bladder, esophagus, colon, breast, stomach, small intestine, lung, liver, and kidney, whereas normal tissues contain either no protein or very little protein. DDX43 has been discovered to have nucleic acid unwinding activity by Yadav et al. and others. Because DDX43 is overexpressed in a wide range of malignancies, making it a potential biomarker or therapeutic target. To predict anthracycline treatment response in breast cancer, expression of DDX43 could be used as a sign and prognostic marker [10,11].

DDX43 was shown to be overexpressed in breast cancer.its overexpression was linked to poor prognosis, aggressive clinical and pathological features in breast cancer patients [12]. Though, the DDX43 expression and its clinicopathological importance within various BC subtypes remain unclear.

Due to the fact that an individual's whole blood transcriptome can reliably forecast the expression levels of about 60% of genes in 32 tissues. The blood transcriptome can predict disease states for various complex diseases almost as well as measured tissue expression. Prognostic markers based on blood transcriptome are being created for a variety of complex diseases, including cancer. **Basu et al.** demonstrated that while the blood transcriptome can aid in biomarker detection in some circumstances, precise models for predicting tissue-specific expression based on the blood transcriptome can be more efficient [13].

As well blood sampling is far less invasive than performing tissue biopsies for prognosis, predicting chemotherapy response or monitoring. However, no prior study has examined its blood levels. therefore, We aim to assess the levels as well as clinicopathological significance of DDX43 expression in different BC subtypes. The current study had the following goals: (a) assess the blood DDX43 protein and mRNA expression levels in benign and malignant BC patients compared to control subjects (b) evaluate the correlation between DDX43 protein levels and mRNA expression levels (c) investigate the possible relationships between both DDX43 protein, mRNA levels and clinicopathological characteristics (d) examine the potential interacting proteins with DDX43 protein via one of the commonly used protein-protein interaction databases.

## Patients and methods

### Ethics statement

The study design was agreed by the Institutional Review Board (IRB) at the Baheya foundation, a nonprofitable BC center in Giza, Egypt. The study protocol was agreed by the Scientific Research Ethical Committee of Faculty of Pharmacy (Girls), Al-Azhar University in compliance with the Declaration of Helsinki's ethical standards., (approval number 368), In addition, before being included in the study, each subject was required to complete a written informed consent form.

### Sample size calculation

We calculated the sample size in G*Power 3.1.9.4 software using the following formula: We selected the ANOVA: Fixed effects, omnibus, one-way from the F-test family with 0.4 effect size, 0.8 power and 5% significance level. This yielded 66 individuals divided between the malignant, benign and control groups. This sample size could not be achieved in the benign group due to the low incidence of benign cases in proportion to malignant cases during recruitment of the cases for the study. The effect size was considered large because there were many previously published studies showing large effect sizes regarding mRNA expression of DDX43 in tumors compared to normal tissues [9–11].

### Study population and specimen collection

An observational case-control study was employed. Random unrelated 80 female patients newly diagnosed with benign and malignant BC aged 27–74 years who presented to the outpatient clinic of Baheya foundation for Early Detection and Treatment of Breast Cancer were recruited for this study between March 2019 and February 2020. Twenty control individuals of

the same age and gender were involved if they had not any clinical symptoms or suspecting data showing a history of illness of BC or any type of cancer.

Inclusion criteria for patients included the diagnosis of BC either malignant or benign. Any concomitant type of cancer other than breast cancer was a criterion for exclusion for both patients and controls.

Based on morphologic analysis of the patient samples, grading of the tumor and histopathologic diagnosis were performed using the World Health Organization categorization of breast cancers, fifth edition (2019). Examining the removed specimens allowed the pathologic stage to be established. The seventh version of the tumor-node-metastasis (TNM) classification by the American Joint Committee on Cancer (AJCC) is used [14]. A senior pathologist evaluated and confirmed all histopathologic data.

Metastatic workups for all patients included a chest radiograph, bone scanning, and abdominal sonar as well as complete history, and clinical examination. True-cut biopsy samples were performed for all patients for histopathological diagnosis and testing of hormone receptors (ER, PR), and HER2 by Ventana BenchMark XT Autostainer). A senior pathologist reviewed the ER, PR, and HER2 data and reported it in accordance with the new guidelines released by the American Society of Clinical Oncology/ the College of American Pathologists in 2018. Thus, ER and PR positivity is defined as 1% to 100% of tumor nuclei positive. Immunohistochemistry (IHC) staining of 3+ (strong and full circumferential membrane staining in > 10% tumor cells) showed HER2-positive (HER2+) tumors. HER2-negative scores are 0 or 1+. In situ hybridization determined HER2 status for 2+ (equivocal) IHC labeling [15].

Molecular subtypes of breast cancer were divided into five groups: luminal A (ER+, PgR + or PgR-, HER2-, and low Ki-67 index), luminal B (HER2 -) (ER+, PgR+ or PgR-, HER2-, and high Ki-67 index), Luminal B (HER2+) (ER+, PgR+ or PgR-, and HER2+), HER2 (ER-, PgR-, and HER2+), and basal-like (ER-, PgR-, and HER2-) [16].

Five milliliters of peripheral blood were drawn from all subjects prior to any treatment, 2ml was collected for RNA extraction into K3 EDTA coated vacuum tubes, while the remaining 3 ml was collected into serum vacuum tubes with clot activator and instantly centrifuged to isolate the serum. Serum samples were split into aliquots and kept at -40°C until the assay.

## Measurement of serum helicase antigen gene (DDX43)

Serum DDX43 levels were measured using a quantitative double-antibody sandwich enzyme-linked immunosorbent (ELISA) assay and a human monoclonal antibody as per the instructions provided by the manufacturer. A commercial kit was provided by Assay Kit Co., USA. Using the standard calibration curve, we determined the DDX43 levels in the serum specimens.

## RNA extraction and cDNA synthesis

Samples of blood were processed using the QIAamp RNA Blood Mini Kit (Qiagen, Germany) to get their total RNA content. A Nanodrop ND-1000 spectrophotometer (Thermo Fisher Scientific, USA) was used to assess RNA concentrations and quality. High-Capacity cDNA Reverse High-Capacity cDNA Reverse Transcription Kit (Applied Biosystems, Thermo Fisher Scientific, USA) was used to reverse-transcribe the RNA to cDNA. Following the manufacturer's instructions, cDNA was synthesized using 10 μl of total RNA.

## Quantitative real-time PCR (q real-time PCR) from blood samples

Custom primers (Thermo Fisher) for the helicase antigen gene (DDX43) (forward, 5´ -GGAGATCGGCCATTGATAGA-3´and reverse, 5´ -GGATTGGGGATAGGTCGTTT-3´ and the

housekeeping gene hypoxanthine phosphoribosyl transferase 1 (HPRT1) (forward, 5′ -TGACACTGGCAAAACAATGCA-3′ and reverse, 5′ -GGTCCTTTTCACCAGCAAGCT-3′ and Maxima SYBR Green/ROX qPCR Master Mix (2X) were then used in quantitative real-time PCR on Applied Biosystems StepOne Real-Time PCR System (Thermo Fisher Scientific Inc, USA). A total of 12.5 μl comprising 2.5 μl cDNA (≤200 ng), 1 μl primer mix (the forward and reverse primers were dissolved in water to 25 pmol/ml concentrations), 0.1 μl ROX solution, 6.25 μl of SYBR Green master mix, and 2.65 μl of water were used in the polymerase chain reaction. The PCR conditions were: an activation step at 95˚C for 10 minutes followed by 40 cycles of denaturation at 95˚C for 15 seconds, annealing at 60˚C for 30 seconds, and extension at 72˚C for 30 seconds. Melt curves analysis was performed. Housekeeping gene values were averaged and calculations were made to normalize the expression of the gene of interest using the comparative CT method ($2^{- \Delta\Delta CT}$ method) [17].

## Statistical analysis

To perform the statistical analysis, we used the Statistical Package for the Social Sciences (SPSS, version 16). The quantitative data was represented as a mean ± standard deviation. The Kolmogorov-Smirnov and Shapiro-Wilk tests were used to test the normality of the data. Non-normally distributed data is normalized by inverse or square root transformation. An independent T-test or one way Analysis of Variance (ANOVA) followed by a post hoc test LSD was utilized in comparing the normally distributed data between two or more groups. To compare mean values that are not normally distributed between two or more groups, the Mann-Whitney U Test or Kruskal-Wallis test was used. Spearman correlation coefficient was used to evaluate the correlation between DDX43 protein levels and mRNA expression levels. P-values less than 0.05 were considered statistically significant, while those between 0.05 and 0.1 were considered marginally significant.

## Results

### Study population characteristics

A total of 100 samples were analyzed, Tables 1 and 2. Eight samples from BC patients and two samples from controls were excluded because results from the quantitative real-time PCR analysis were undetermined. In control, benign as well as malignant cases, the blood levels of DDX43 protein and DDX43 mRNA expression were assessed. Although the mean normalized serum levels of DDX43 protein were slightly higher in control than in both benign and malignant groups, this difference was non-significant (p = 0.77, p = 0.68). The mean normalized level of DDX43 mRNA expression was 1.5-times higher in control than in benign cases, but this result was marginally significant (p = 0.14). Also, the mean normalized level of DDX43 mRNA expression was three-times higher in control than in malignant cases, while this result was not statistically significant (p = 0.51). However, in benign cases compared malignant cases, the mean normalized level of DDX43 mRNA expression was significantly higher, (p = 0.016). Also, using non-parametric test (Mann-Whitney test), the mean level of DDX43 mRNA expression was significantly higher in benign cases compared to malignant cases (p = 0.03), (Fig 1). The Nottingham Prognostic Index Plus (NPI+) for the biological classes was calculated according to NPI+ formulae for the biological classes (Table 3).

In all subjects, control, benign, and malignant groups there were insignificant weak correlations between DDX43 protein serum levels and mRNA expression blood levels (r = -0.079, r = -0.089, r = 0.095, and r = -0.083 respectively), (Fig 2).

**Table 1. Demographic and clinical characteristics of studied groups (n = 90).**

| Characteristics | Control gp (n = 18) | Benign gp (n = 12) | Malignant gp (n = 60) | P-value |
|---|---|---|---|---|
| **Age** (years) (mean ± SD) | 48.94±10.42 | | 52.07±10.24 | 0.251 |
| | 48.94±10.42 | 36.09±8.10 | 52.07±10.24 | 0.001 |
| | | 36.09±8.10 | | 0.0001 |
| **Body mass index** (BMI), kg/m² (mean ± SD) | 32.62±7.23 | | 34.48±6.44 | 0.317 |
| | 32.62±7.23 | 27.94±3.49 | 34.48±6.44 | 0.347 |
| | | 27.94±3.49 | | 0.171 |
| **Family History of Cancer n (%)** | | | | |
| Yes | 14 (77.8%) | 9 (75%) | 41 (68.3%) | 0.704 |
| No | 4 (22.2%) | 3 (25%) | 19 (31.7%%) | |
| **Diabetes/Diabetes medications n (%)** | | | | |
| Yes | 7 (38.9%) | 4 (33.3%) | 18 (30%) | 0.775 |
| No | 11 (61.1%) | 8 (66.7%) | 42 (70%) | |
| **Menopausal status** n (%) | | | | |
| Pre | 7 (38.9%) | 10 (83.3%) | 26 (43.3%) | 0.028 |
| Post | 11 (61.1) | 2 (16.7%) | 34 (56.7%) | |
| **mRNA expression of DDX43** (mean ± SD) | 3.33±6.64 | | 1.47±1.75 | 0.507 |
| | 3.33±6.64 | 3.1±3.57 | 1.47±1.75 | 0.119 |
| | | 3.1±3.57 | | 0.018 |
| **Serum DDX43 protein levels** (ng/L) (mean ± SD) | 2148.4±2025.69 | | 1905.8±1740.05 | 0.683 |
| | 2148.4±2025.69 | 1902.3±1401.47 | 1905.8±1740.05 | 0.764 |
| | | 1902.3±1401.47 | | 0.989 |

## Serum DDX43 protein levels, mRNA expression of DDX43 in relation to clinicopathological parameters

We then compared the serum DDX43 protein levels and its mRNA expression in the malignant breast cancer patients in subgroups according to the pathological parameters, Table 4.

The mean normalized serum level of DDX43 protein was significantly lower in the IDC tumor type than in both ILC and ICC tumor types (p = 0.042, p = 0.006). The mean normalized serum level of DDX43 protein was higher in patients with tumor size (T2) than tumor size (T1), but the result was marginally significant (p = 0.058). Whereas the mean normalized serum level of DDX43 protein was significantly lower in patients with tumor size (T3-4) than tumor size (T2), (p = 0.039). While the mean normalized serum level of DDX43 protein was lower in the ILC tumor type than in the ICC tumor type but the result was marginally significant (p = 0.152). Also, the mean normalized serum level of DDX43 protein was higher in the mitosis score 2 than the mitosis score 3, but the result was marginally significant (p = 0.137). Similarly, the mean normalized serum level of DDX43 protein was higher in nuclear grade 2 than nuclear grade 3, but the result was marginally significant (p = 0.088). There were no significant differences between the mean normalized serum DDX43 protein levels in different groups based on body mass index, menopausal status, tumor size (T1 vs. T3-4), tumor-node-metastasis (TNM) stage, lymph node stage, mitosis score (1 vs. either 2 or 3), tubular formation score, histological grade, tumor grade, hormonal receptors, HER2 Family, triple-negative phenotype, NPI, biological classes and NPI+ for the biological classes (Results are not shown).

The mean normalized level of DDX43 mRNA expression was considerably higher in tumor grade 3 than tumor grade 2, (p = 0.037). In addition, in tumor grade 3 the mean normalized level of DDX43 mRNA expression was higher than in tumor grade 1, but the result was marginally significant (p = 0.056). While the mean normalized level of DDX43 mRNA expression was lower in mitosis score 1 than in both mitosis scores 2 &3. But the result was marginally significant (p = 0.129, p = 0.108). Also, the mean normalized level of DDX43 mRNA expression

**Table 2. Pathological features.**

### A) Pathological features of malignant group

| Characteristics Malignant gp (n = 60( | | | |
|---|---|---|---|
| **Tumor size (continuous)** n (%) | T1 (<2 cm) 10 (16.67%) | T2 (2–5 cm) 22 (36.66%) | T3-4 (>5 cm) 27 (45%) | NA 1 (1.67%) |
| **Lymph node stage** n (%) | Negative 28 (46.66%) | Positive (1–3 nodes) 16 (26.67%) | Positive (>3 nodes) 15 (25%) | NA 1 (1.67%) |
| **Mitosis score** n (%) | 1 7 (11.67%) | 2 44 (73.33%) | 3 9 (15%) | |
| **Tubular formation score** n (%) | 1 2 (3.33%) | 2 27 (45%) | 3 26 (43.34%) | NA 5 (8.33%) |
| **Nuclear grade** n (%) | 1 4 (6.67%) | 2 40 (66.67%) | 3 16 (26.67%) | |
| **Histological grade** n (%) | 1 4 (6.67%) | 2 36 (60%) | 3 16 (26.67%) | NA 4 (6.66%) |
| **Tumor grade** n (%) | Grade 1 (low) 7 (11.67%) | Grade 2 (intermediate) 36 (60%) | Grade 3 (high) 17 (28.33%) | |
| **Hormonal receptors** Estrogen receptor n (%) | Negative 9 (15%) | Positive 51 (85%) | | |
| Progesterone receptor n (%) | Negative 10 (16.67%) | Positive 49 (81.67%) | NA 1 (1.66%) | |
| **Proliferation/cell cycle regulators** Ki67 LI n (%) | Low (immunostaining <20%) 4 (6.67%) | High (immunostaining ≥20%) 12 (20%) | NA 44 (73.33%) | |
| **HER2 Family** n (%) HER2 | Negative 52 (86.67%) | Positive 6 (10%) | NA 2 (3.33%) | |
| **Triple-negative phenotype** n (%) | No 53 (88.33%) | Yes 7 (11.67%) | | |
| **Nottingham prognostic index (NPI)** n (%) | Good prognostic group (≤3.4) 10 (16.67%) | Moderate group (3.41–5.4) 31 (51.66%) | Poor group (>5.41) 13 (21.67%) | NA 6 (10%) |
| **NPI+ for the biological classes** n (%) | Good prognostic group (≤3.4) 30 (50%) | Moderate group (3.41–5.4) 14 (23.33%) | Poor group (>5.41) 1 (1.67%) | NA 15 (25%) |
| **Chemotherapy** n (%) | Yes (after blood sampling) 38 (63.33%) | No 22 (36.67%) | | |
| **Peritumoural lymphovascular invasion** | absent 41 (68.34%) | suspicious 6 (10%) | present 8 (13.33%) | NA 5 (8.33%) |
| **DCIS in specimen** | absent 28 (46. 67%) | present 32 (53.33%) | | |
| **Microcalcification** | absent 42 (70%) | present 18 (30%) | | |

### B) Pathological features of benign and malignant groups

| Characteristics | Benign gp (n = 12) | Malignant gp (n = 60) |
|---|---|---|
| **Tumor-node-metastasis (TNM) stage** n (%) Stage 0—Tis, N0, M0. Stage I - T1-T2, N0, M0. Stage II - T2-T4, N0, M0. Stage III - T1-T4, N1-N3, M0. Stage IV - T1-T4, N1-N3, M1. | | 2 (3.33%) 27 (45%) 0 (0%) 31 (40.67%) 0 (0%) |

(*Continued*)

| **Tumor type n (%)** | | |
| --- | --- | --- |
| Invasive duct carcinoma (IDC) | 6 (50%) | 36 (60%) |
| Invasive lobular carcinoma (ILC) | 2 (16.7%) | 8 (13.3%) |
| Invasive Cribriform carcinoma (ICC) | 4 (33.3%) | 5 (8.3%) |
| Mixed | | 4 (6.7%) |
| Ductal carcinoma in situ (DCIS) | | 3 (5%) |
| Others | | 4 (6.7%) |
| Fibroadenomas | | |
| Phyllodes Tumors | | |
| Benign others | | |
| **Location** n (%) | | |
| Right | 6 (50%) | 27 (45%) |
| Left | 6 (50%) | 32 (53.3%) |
| Bilateral | | 1 (1.7%) |
| **Biological classes** | | |
| Luminal A | | 29 (48.3%) |
| Luminal B | | 20 (33.3%) |
| Luminal N | | 8 (13.4) |
| HER2+/ER− | | 1 (1.7) |
| NA | | 2 (3.3%) |

NA: Not available.

Tumor-node-metastasis (TNM) stage: Stage 0—Indicates carcinoma in situ. Tis, N0, M0, Stage I—Localized cancer. T1-T2, N0, M0, Stage II—Locally advanced cancer, early stages. T2-T4, N0, M0, Stage III—Locally advanced cancer, late stages. T1-T4, N1-N3, M0, Stage IV—Metastatic cancer. T1-T4, N1-N3, M1 [18]. The NPI was calculated using the following formula: NPI = histological grade (1–3) +LN stage (1–3; 1 = negative, 2 = 1–3 nodes positive, 3 = ≥4 nodes positive) + (tumour size/ cm × 0.2) [19].

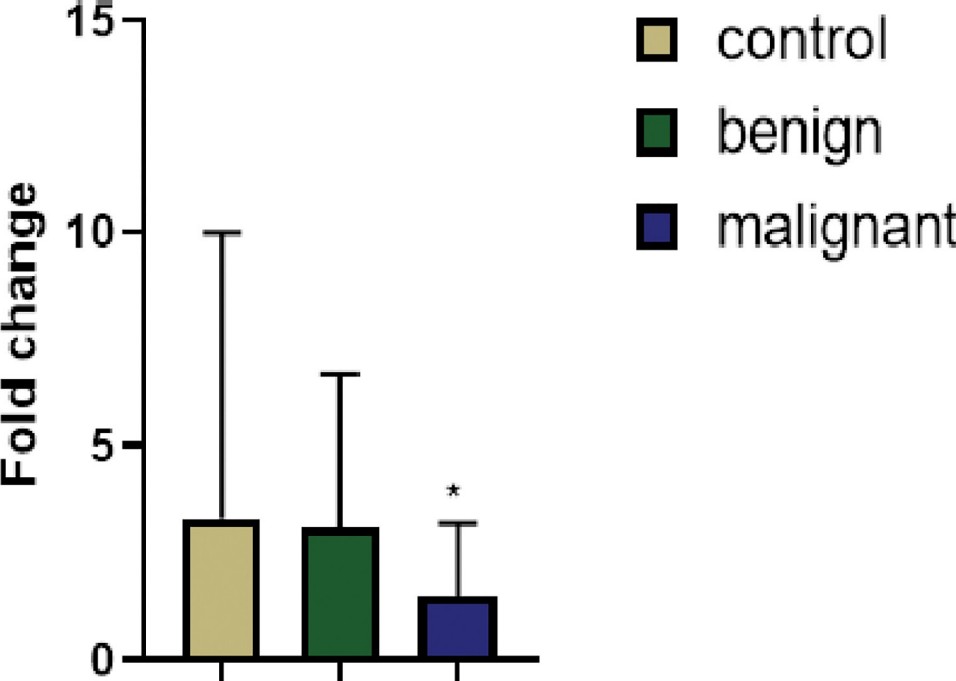

**Fig 1. Fold change expression of the DDX43 gene relative to the reference gene (HPRT1), relative to the expression in the control samples.** Bar heights indicate the mean expression of the gene in several samples in the studied groups. Error bars show the standard deviation of the fold changes in each group. One asterisk indicates a statistically significant difference between the means of the malignant group compared to the mean of the benign group.

**Table 3. NPI+ formulae for the biological classes [20].**

| Class | NPI+ formula |
|---|---|
| Luminal A | $(0.8 \times Mitosis) + (0.5 \times Size) + (1.8 \times Nodal\ ratio*)$ |
| Luminal N | $(0.8 \times Tubules) + (0.6 \times Stage)$ |
| Luminal B | $(0.7 \times Mitosis) + (1.0 \times Nodal\ ratio)$ |
| HER2+/ER+ | $(0.5 \times Size) + (0.9 \times Stage)$ |
| HER2+/ER− | $(0.9 \times Stage) − (0.6 \times Nodal\ ratio)$ |

* Number of nodes positive/Total number of nodes.

was higher in nuclear grade 3 than nuclear grade 2, but the result was marginally significant (p = 0.085). As well, the mean normalized level of DDX43 mRNA expression was lower in patients with tumor size (T2) and (T3-4) than tumor size (T1), but the result was marginally significant (p = 0.110, p = 0.141). The mean normalized level of DDX43 mRNA expression was lower in estrogen receptor-positive patients than in estrogen receptor-negative patients. But the result was marginally significant (p = 0.168). Whereas the mean normalized DDX43 mRNA expression level was higher in patients with triple-negative phenotypes than in other patients, the result was marginally significant (p = 0.187). There were no significant differences between the mean normalized levels of DDX43 mRNA expression in different groups based on body mass index, menopausal status, tumor size (T2 vs. T3-4), TNM stage, lymph node stage, tumor type, mitosis score (2 vs. 3), tubular formation score, histological grade, tumor grade (grade 1 vs. grade 2), hormonal receptors (progesterone receptor negative vs. positive), HER2 Family, NPI, biological classes and NPI+ for the biological classes (Results are not shown).

DDX43 was predicted to have three protein interacting networks (PINs) using the STRING data (https://stringdb.org/cgi/)(human database) under high confidence (with a minimal interaction score of 0.700), and the results showed three functional partners: Sarcoma antigen 1 (SAGE1), Melanoma-associated antigen 1 (MAGEA1), and a probable ATP-dependent RNA-helicase; DEAD-box helicase 53 (DDX53) [21].

Correlation networks for these four proteins were built using carefully selected and experimentally verified data. The "more" button on the STRING interface was used to add an additional 20 nodes/proteins to the initial network of three proteins. The functional partner proteins with DDX43 are listed in the table, and the confidence cutoff value for interaction linkages has been adapted to 0.700, (Fig 3A), and supplementary table (Fig 3B). As a consequence of the above findings, we used Kyoto Encyclopedia of Genes and Genomes (KEGG) mapping to map DDX43 protein to molecular interaction/reaction/relation networks (KEGG pathway maps, BRITE hierarchies, and KEGG modules) revealed no pathways mediated by DDX43. This shows that there is no evidence for such pathways. KEGG pathway is based on curated and verified information. So, such information is not proven.

## Discussion

DDX43 is an ATP-dependent dual helicase that relaxes both DNA and RNA molecules, according to **Talwar et al.** This demonstrates its putative cellular roles, which may be linked to its overexpression in malignancies. In the presence of Mg2+ and ATP, it catalyzes the unwinding process most effectively [22].

While the DDX43 gene has been proposed as a possible oncogene because of its high expression in various types of cancer, details about its precise physical function are scarce.

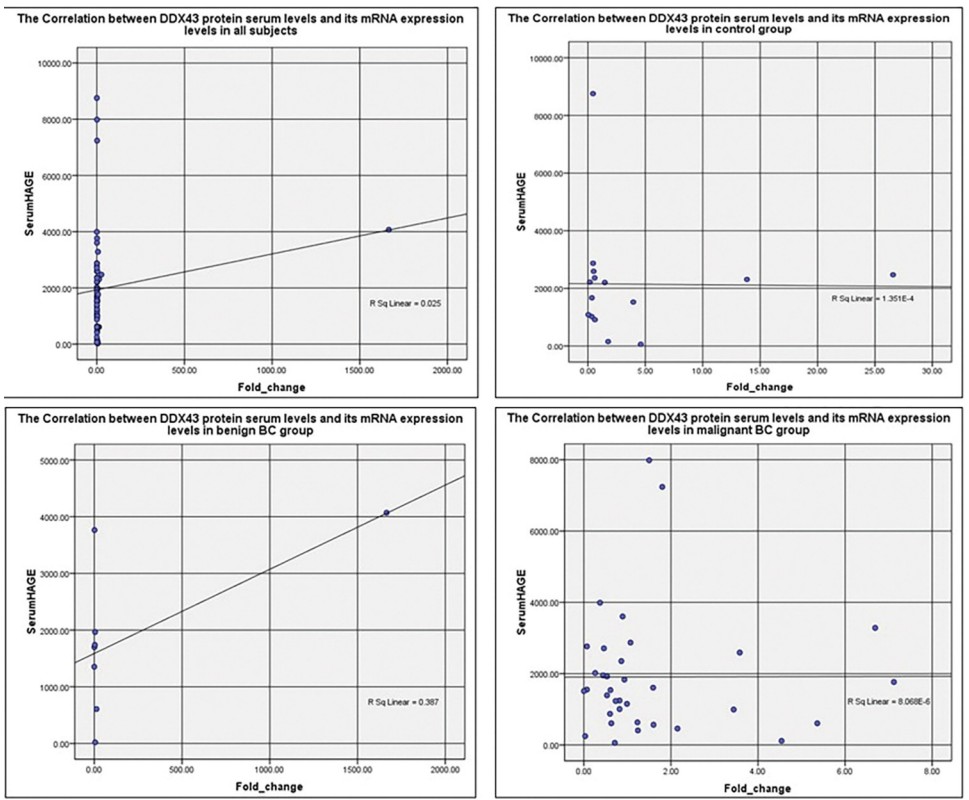

**Fig 2. Scatter plot representation of the correlation between DDX43 protein serum levels and its mRNA expression levels in all subjects, control, benign, and malignant groups.**

## Serum protein levels and mRNA expression of DDX43 in the studied groups

The mean normalized serum DDX43 protein level was slightly higher in the control than in both benign and malignant groups, but this result was non-significant. To our knowledge, there is little information about DDX43 protein levels in serum. This may be as it is not predicted to be definitely secreted to blood and thus not been analyzed [23]. However, the results regarding serum DDX43 level were somewhat similar to a previous study. DDX43 protein was found in a range of tumor tissues, which include the brain, bladder, esophagus, colon, breast, stomach, small intestine, lung, liver, and kidney, while not in normal tissues or at extremely low levels, according to this study. It showed low staining of DDX43 protein in breast cancer tissues like serum DDX43 protein level in malignant breast cancer patients stated in the current study [24]. Also, the human atlas protein shows that DDX43 antibody staining is not detected in 9 of 11 human breast cancer tissue, detected low in 1 of 11 and medium in 1 of 11 [25].

As we know, DDX43 mRNA expression in breast cancer patients has been studied before. We have found that DDX43 expression in the blood of malignant breast cancer patients is under-expressed. Unlike other studies that have found it overexpressed in a number of solid tumors, including the brain, salivary gland, lung, colon, and prostate cancers, as well as hematologic malignancies (e.g., chronic myeloid leukemia and multiple myeloma) [26,27]. These studies, however, examined DDX43 mRNA expression in malignant tissues not in blood.

**Table 4. Serum DDX43 protein levels and its mRNA expression in malignant breast cancer patients subgroups according to some pathological parameters.**

| Characteristics | Serum DDX43 protein levels (ng/L) (mean ± SD) | P-value | mRNA expression of DDX43 (mean ± SD) | P-value |
|---|---|---|---|---|
| **Pathological parameters** | | | | |
| **Tumor size (continuous)** (n) <br> T1 (<2 cm) (10) vs. T2 (2–5 cm) (22) <br> T2 (2–5 cm) (22) vs. T3-4 (>5 cm) (27) <br> T1 (<2 cm) (10) vs. T3-4 (>5 cm) (27) | 1294±1024.64 <br> 2783.5±2317.01 <br> 2783.5±2317.01* <br> 1333.9±825.99 | 0.058 <br> 0.039 | 2.28±2.14 <br> 1.24±1.57 <br> 2.28±2.14 <br> 1.40±1.74 | 0.110 <br> 0.141 |
| **Tumor type** (n) <br> IDC (36) vs. ILC (8) <br> IDC (36) vs. ICC (5) <br> ILC (8) vs. ICC (5) | 1177±934.84* <br> 2103.1±936.43 <br> 1177±934.84* <br> 4955±4284.34 <br> 2103.1±936.43 <br> 4955±4284.34 | 0.042 <br> 0.006 <br> 0.152 | | |
| **Mitosis score** (n) <br> 1 (7) vs. 2 (44) <br> 2 (44) vs. 3 (9) <br> 1 (7) vs. 3 (9) | 2271.8±1948.29 <br> 1142.1±870.3 | 0.137 | 0.67±0.88 <br> 1.67±1.93 <br> 0.67±0.88 <br> 1.29±1.34 | 0.129 <br> 0.108 |
| **Nuclear grade** (n) <br> 2 (40) <br> 3 (16) | 2252.1±1944.33 <br> 1273.1±1167.98 | 0.088 | 1.22±1.36 <br> 2.3±2.55 | 0.085 |
| **Tumor grade** (n) <br> Grade 2 (36) vs. grade 3 (17) <br> Grade 1 (7) vs. grade 3 (17) | | | 1.3±1.62 <br> 2.06±2.17* <br> 0.86±0.82 <br> 2.06±2.17 | 0.037 <br> 0.056 |
| **Hormonal receptors** <br> Estrogen receptor (n) <br> Negative (9) <br> Positive (51) | | | 2.07±2.21 <br> 1.36±1.66 | 0.168 |
| **Triple-negative phenotype** (n) <br> No (53) <br> Yes (7) | | | 1.35±1.63 <br> 2.36±2.47 | 0.187 |

Regarding the results, there is compatibility between DDX43 protein and mRNA expression blood levels in both the control, benign and malignant groups (Both protein and mRNA expression blood levels are higher in control than in both benign and malignant groups).

## Serum protein levels and mRNA expression of DDX43 in relation to clinicopathological parameters

The mean normalized serum DDX43 protein level was significantly lower in the IDC tumor type than in both ILC and ICC tumor types.

This is consistent with **Mathieu et al.** who reported little protein level in situ expression of DDX43 in breast cancer tissue (invasive ductal carcinoma) [24].

In various studies comparing clinical and pathological aspects, metastatic sites, and survival rates of IDC and ILC, most of the results were contradictory. Nevertheless, most research found no significant difference regarding most clinicopathological features, treatment aspects, as well as survival between the IDC and ILC groups. But regarding vascular invasion, it was lower in ILC than in IDC and the difference was significant [28]. This means that IDC has a poor prognosis than ILC.

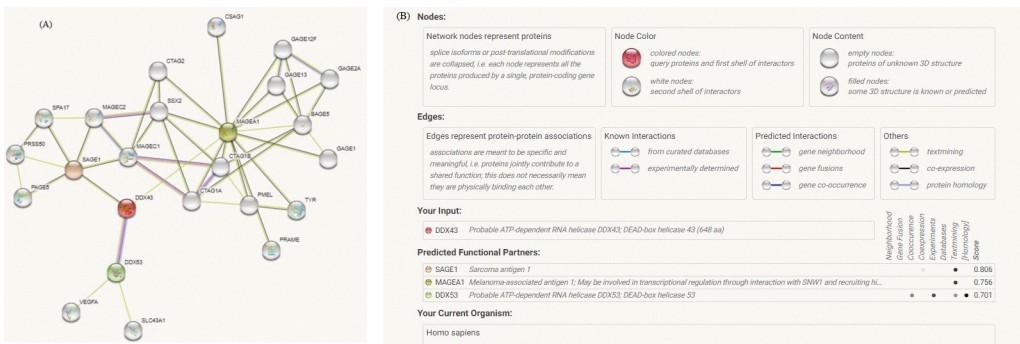

**Fig 3. STRING database predicted protein interacting networks (PINs) of DDX43.** (DDX43, Probable ATP-dependent RNA helicase DDX43; DEAD-box helicase 43). (**A**) The network began with three proteins and was moreover joined by an additional 20 nodes/proteins, by using the "more" key on the STRING interface. The network nodes are proteins. The edges, which represent the predicted functional associations, may be drawn with up to 7 differently colored lines representing the existence of the seven types of evidence used in predicting the associations. Red line—indicates fusion evidence. Green line—neighborhood evidence. Blue line—cooccurrence evidence. Purple line—experimental evidence. Yellow line—textmining evidence. Light blue line—database evidence. Black line—coexpression evidence. Line thickness indicates the strength of data support. (Abbreviations: SAGE1: Sarcoma antigen 1, MAGEA1: Melanoma-associated antigen 1, DDX53: Probable ATP-dependent RNA helicase; DEAD-box helicase 53, VEGFA: Vascular endothelial growth factor A, SLC43A1: Large neutral amino acids transporter small subunit 3; Sodium-independent, CTAG1A: Cancer/testis antigen 1A, CTAG1B: Cancer/testis antigen 1B, CTAG2: Cancer/testis antigen 2, PRAME: PReferentially expressed Antigen in Melanoma, PMEL: Melanocyte protein, BAGE5: B melanoma antigen 2, TYR: Tyrosinase, GAGE1: Cancer-associated gene 1 protein; Cancer antigen 1, GAGE12F: G antigen 12F, GAGE2A: G antigen 2A, GAGE13: G antigen 13, CSAG1: Putative chondrosarcoma-associated gene 1 protein, MAGEC1: Melanoma-associated antigen C1, MAGEC2: Melanoma-associated antigen C2, SPA17: Sperm surface protein Sp17, PAGE5: Prostate-associated gene 5 protein; P antigen family member 5, SSX2: Synovial sarcoma, X breakpoint 2, PRSS50: Probable threonine protease.) (B) The supplementary table displays all interacting proteins with DDX43, and the confidence cutoff value for interaction linkages has been adapted to 0.700 as the high-scoring link.

Although both ICC and IDC belong to a family of low-grade breast cancer, **Zhang et al.** found that ICC was related to a number of advantageous prognostic aspects than IDC [29]. This proposes that lower serum DDX43 protein levels may be linked to types of breast cancer with a bad prognosis or with a poor prognosis.

The mean normalized serum DDX43 protein level was higher in patients with tumor size (T2) than tumor size (T1), but the result was marginally significant. Whereas the mean normalized serum level of DDX43 protein was significantly lower in patients with tumor size (T3-4) than in tumor size (T2).

In cases of broad lymph node involvement, extremely small tumors, according to **Wo et al.**, may represent a more aggressive subtype than larger tumors with equivalent lymph node contribution [30]. Thus, tumor size T1 may correlate to poor clinical outcome or worse prognosis which could explain the lower serum DDX43 protein level.

Also, the mean normalized serum DDX43 protein level was higher in mitosis score 2 than mitosis score 3, but the result was marginally significant.

According to numerous studies, the mitotic count is a critical element in determining histological grade. Regardless of the technique utilized, **Van Diest et al.** noted that higher proliferation is closely associated with a bad prognosis [31]. This also supports that low serum DDX43 protein level is associated with mitosis score with worse prognosis.

Similarly, the mean normalized serum DDX43 protein level was higher in nuclear grade 2 than in nuclear grade 3, but the result was marginally significant.

In breast cancer, the nuclear grade is an additionally known prognostic component. In **Yang et al.** study, nuclear scoring was linked positively with proliferative activity. Also, nuclear pleomorphism, cell differentiation, and mitotic frequency are all aspects of cell morphology

that are included in nuclear grade [32]. These studies, together with the present study results, support that serum DDX43 protein level is negatively associated with the more aggressive the tumor cells tend to be.

The mean normalized level of DDX43 mRNA expression was greater in tumor grade 3 than in tumor grades 2 and 1. The result was statistically significant regarding grade 2 vs. grade 3, but it was marginally significant regarding grade 1 vs. grade 3.

**Schwartz et al.** examination shows that the histologic grade in breast cancer is still a predictive factor despite alterations in tumor size and the number of positive lymph nodes. As the histologic grade increased from G1 to G3 for each grouping of T and N, the survival rates progressively reduced as the tumor size as well as the number of entailed nodes increased for every grade level [33]. This supports that higher DDX43 mRNA expression in some malignant BC patients is linked to higher tumor grades.

While the mean normalized level of DDX43 mRNA expression was lower in mitosis score 1 than in both mitosis scores 2 &3. But the result was marginally significant. Also, the mean normalized level of DDX43 mRNA expression was higher in nuclear grade 3 than in nuclear grade 2, but the result was marginally significant. As well, the mean normalized level of DDX43 mRNA expression was lower in patients with tumor size (T2) and (T3-4) than in tumor size (T1), but the result was marginally significant.

These results are also compatible with DDX43 mRNA expression in relation to tumor grade. All support that higher DDX43 mRNA expression may be linked to higher mitosis scores, higher nuclear grades, and smaller tumor size.

They agree with **Abdel-Fatah et al.** finding that DDX43+ expression is considerably linked to aggressive clinicopathological and high proliferation metrics and is also a new independent predictor of poor prognosis [34].

The mean normalized levels of DDX43 mRNA expression were lower in the estrogen receptor-positive patients than in estrogen receptor-negative patients, but the result was marginally significant. Whereas the mean normalized levels of DDX43 mRNA expression were higher in patients with triple-negative phenotypes, the result was marginally significant.

Both results are compatible with what **Abdel-Fatah et al.** study indicated. Their study stated that DDX43+expression was related to ER-negative expression along with other aggressive features, such as HER2 gene overexpression, TNBC, high grade, and proliferation [34].

According to the patent (WO2013144616), increased expression of DDX43 correlates substantially with aggressive clinicopathological parameters, as high proliferation, absence of concurrent expression of ER, progesterone receptor, and HER2 (triple negative), as well as overexpression of both epidermal growth factor receptor (EGFR) and HER2. Considering the restrictions of the present chemotherapy, the inventors state moreover DDX43 as a likely target to treat cancer with a DDX43-specific chemotherapeutic agent antigen or DDX43-specific antibody. Therefore, the DDX43 level can function as a biomarker in the outline of the contemporary diagnosis as well as pharmacological therapy of patients with BC [35].

Since breast cancer patients have lower blood levels of DDX43 gene expression relative to normal subjects, this may be due to DDX43 gene mutation in breast cancer. This mutation needs further investigation to reveal it. Other than regulating translation, abnormal DDX43 expression could play a role in oncogenesis. Therefore, more investigation is required to better understand the role of DDX43 in both translation start and oncogenesis.

The noticeable inconsistency between the behaviors claimed DDX43 to promote tumor growth versus the present study results that revealed higher DDX43 protein and mRNA expression blood levels in both control and benign groups than in malignant group might be explained in a variety of ways. Mostly, this is because all these studies assessed DDX43 protein expression in tumor tissues, not in blood or serum. Also, **Abdel-Fatah et al.** in that study

stated high HAGE expression (HAGE+) in only 8% of tumors from early primary BC (EP-BC) patients [34]. In this context, the results of **Oloomi et al.** study were the same as ours. While there were significant differences for some molecular markers in blood samples between healthy individuals and those with breast cancer, these differences were not present in tissue samples [36]. Additionally, many of the earlier DDX43 investigations were focused on the cellular functional link, while DDX43 expression and signaling pathways have not been explored. So comprehensive expression-cellular impact association studies are mandatory to establish their functional link.

Although the **Bourgeois et al.** review stated that DDX43 is one of the RNA helicases that are engaged in translation activation, they found that their exact roles remain equivocal [37].

Another explanation for the down-regulation of DDX43 gene expression in the blood of breast cancer patients may be attributable to low $Mg^{2+}$ levels in the selected cancer patients. Several studies stated that the pathophysiology of numerous diseases involving cancer is caused by an imbalance of magnesium homeostasis. New interesting studies link magnesium, as well as $Mg^{2+}$ transporters, to characteristic and complementary abilities that allow tumor development and metastatic propagation. This has supported the crucial concept of magnesium as a main controller of cell proliferation [38]. Low $Mg^{2+}$ levels may affect DDX43 gene expression, as in the presence of Mg2+, it most efficiently catalyzes the unwinding reaction. However, this hypothesis requires investigating Mg2+ levels as well as the related factors.

## Protein interacting networks (PINs) of DDX43

As proteins have a major role in biological function, their interactions establish cellular and molecular mechanisms that regulate healthy as well as diseased states in organisms. This sequentially can evaluate approaches for prevention, diagnosis, and treatment. Protein networks are beneficial resources to recognize novel pathways to acquire basic information about diseases. Disease-associated interaction proteins can help in identifying possible disease-associated genes and therapeutic targets. Cancer is a complicated disease, and several genes have been stated to entail the development of cancers. A systematic examination of cancer proteins in the human protein-protein interaction (PPI) network might supply major biological information for revealing the molecular mechanisms of cancer and, potentially, other complex diseases [39,40].

STRING database predictions revealed three proteins with functional links to DDX43: SAGE1: Sarcoma antigen 1, MAGEA1: Melanoma-associated antigen 1 and DDX53: Probable ATP-dependent RNA helicase; DEAD-box helicase 53. The evidence pointing to a functional connection between DDX43 and SAGE1 includes co-expression in other organisms and co-mention in Pubmed abstracts. The evidence pointing to a functional relationship between DDX43 and MAGEA1 includes co-mention in Pubmed Abstracts. While the evidence suggesting a functional link between DDX43 and DDX53 includes co-occurrence across genomes, experimental/biochemical data, and co-mention in Pubmed abstracts [21].

**Maheswaran et al.** demonstrated that in normal tissues, the SAGE1 antigen displays a germ cell-specific expression pattern, but was not expressed in breast and lung cancer. This result proposed that SAGE1 cancer/testis antigen is not a hopeful target for breast and lung cancer immunotherapy [41]. There are few studies that report SAGE1 expression in some types of cancer, but there are no studies that examine its role /function in these types or in breast cancer.

Several MAGE proteins are unusually expressed in many cancer types. **Zhao et al.** study showed that MAGEA1 expression is low or undetectable in several breast cancer cell lines ([42]. This study's results resemble a great degree the current results regarding DDX43 under-expression in BC patients compared to normal subjects.

Also, the expression of MAGEA1 is common in TNBC, according to **Raghavendra et al.** [43]. This is similar to DDX43 expression in this type of BC which supports common pathways and mechanisms regulating this type of BC.

Earlier explorations have shown that definite RNA helicases including DDX53 (CAGE) are involved in tumor cell progress and proliferation in numerous types of malignancies [44].

According to reports, DDX43 interacts with DDX53 and DHX15. These findings imply that their interaction partners and the tumor or background environment affect the exact activity of cancer-related DEAD-box proteins [45]. In addition, two studies by **Kim et al.** disclosed that the MDA-MB-231 cells express more DDX53 than MCF-7 cells [46,47]. This result is somewhat similar to ours regarding higher DDX43 expression in patients with triple-negative phenotypes. According to **Singh et al.** phylogenetic analysis further showed that the DDX43 and DDX53 helicases are closely related to one another and likely descended from the same progenitor. BLAST (Basic Local Alignment Search Tool) alignment of DDX43 and DDX53 sequences revealed 62% sequence similarity [48]. This suggests that both DDX43 and DDX53 may possess functional similarity.

The limitations of the present investigation comprised the small sample size, specially of the benign group, as mentioned in the sample size calculation. This could have contributed to the lack of a statistically significant relationship between DDX43 and certain clinicopathological parameters. The small sample size may have affected many of the results that came close to statistical significance. Due to the difficulty of following up with most patients for extended periods of time to determine cancer-specific mortality and/or deaths from any cause over long periods of time, neither cancer-specific survival nor relative survival rates were estimated. To validate our findings, large-scale cohort studies are necessary. Finally, the possible protein interacting networks of DDX43 together with the underlying molecular mechanisms in breast cancer were elucidated using bioinformatics analyses only. As a result, additional in vitro and in vivo research studies are needed to validate the findings of this study.

## Conclusions

In contrast to the documented overexpression of DDX43 in breast cancer tissues and other cancer types, the current investigation found that DDX43 is probably under expressed in the blood of malignant BC patients compared to benign BC and control subjects. DDX43 protein serum levels may be not a reliable and informative assay in differentiating malignant BC patients from either benign or healthy individuals. However, low DDX43 protein serum levels in some malignant BC patients may be related to BC types with bad prognoses. On the other hand, high DDX43 mRNA expression levels in these patients may be linked to the higher tumor and nuclear grades or aggressive types of breast cancer as TNBC. Thus, DDX43 mRNA expression or serum protein levels, or both have the potential to be used as markers of disease progression in human breast cancer. DDX43 mRNA expression was able to effectively discriminate between benign and malignant BC. This proposes a less invasive method for breast cancer follow-up and monitoring. More immunohistochemistry studies, larger validation studies, in vitro and in vivo tests are needed to validate the influence of DDX43 expression on human breast cancer pathogenesis and its worth in prognosis and gene therapy. To examine the role and signaling pathways of DDX43 in breast cancer cells, genetic, biochemical, and cell biology studies should be carried out.

## Acknowledgments

The Authors wish to thank Baheya foundation for early detection and treatment of Breast Cancer for supporting this work.

## Author Contributions

**Conceptualization:** Noha N. Amer.

**Data curation:** Noha N. Amer, Mahmoud M. Kamel.

**Formal analysis:** Noha N. Amer.

**Investigation:** Noha N. Amer, Rabab Khairat.

**Methodology:** Rabab Khairat.

**Resources:** Amal M. Hammad.

**Supervision:** Mahmoud M. Kamel.

**Writing – original draft:** Noha N. Amer.

**Writing – review & editing:** Noha N. Amer, Amal M. Hammad.

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
