## [Decision Letter · Decision Letter 0]

7 Feb 2023

PONE-D-22-26586DDX43 mRNA expression and protein levels in relation to clinicopathological profile of breast cancerPLOS ONE

Dear Dr. Noha Ameir,

Thank you for submitting your manuscript to PLOS ONE. After careful consideration, we feel that it has merit but does not fully meet PLOS ONE’s publication criteria as it currently stands. Therefore, we invite you to submit a revised version of the manuscript that addresses the major points raised during the review process as you will read below this letter.

When responding to the raised points, please be sure to improve the following; introduction  (with proper citations), your justification for the study, methods (including histopathology, HE, IHC, inclusion vs exclusion criteria, sample size calculation etc), statistical analysis, especially  inferential/correlation analysis using proper methods etc) and proper discussion of the findings and conclusion.

We look forward to receiving your revised manuscript.

Kind regards,

Elingarami Sauli, PhD

Academic Editor

PLOS ONE

Journal Requirements:

3. We noted in your submission details that a portion of your manuscript may have been presented or published elsewhere. Please clarify whether this conference proceeding or publication was peer-reviewed and formally published. If this work was previously peer-reviewed and published, in the cover letter please provide the reason that this work does not constitute dual publication and should be included in the current manuscript.

4. Please ensure that you include a title page within your main document. You should list all authors and all affiliations as per our author instructions and clearly indicate the corresponding author.

5. Please include your tables as part of your main manuscript and remove the individual files. Please note that supplementary tables (should remain/ be uploaded) as separate "supporting information" files.

Reviewers' comments:

Reviewer's Responses to Questions

**Comments to the Author**

1. Is the manuscript technically sound, and do the data support the conclusions?

Reviewer #1: No

Reviewer #2: Yes

Reviewer #3: Yes

2. Has the statistical analysis been performed appropriately and rigorously? 

Reviewer #1: No

Reviewer #2: Yes

Reviewer #3: Yes

3. Have the authors made all data underlying the findings in their manuscript fully available?

Reviewer #1: No

Reviewer #2: Yes

Reviewer #3: Yes

4. Is the manuscript presented in an intelligible fashion and written in standard English?

Reviewer #1: No

Reviewer #2: Yes

Reviewer #3: No

5. Review Comments to the Author

Reviewer #1: In this study, the authors evaluated expression levels of DDX43 in breast cancer cases. There some potential points that the authors should consider that;

1) The sample size is too small to conclude this conclusions!

2) The is no correlation among gene or protein expression of DDX43 and clinical data!

3) Figure 3 is irrelevant.

4) The authors should clarify the rational behind the selection of DDX43 gene.

Reviewer #2: the authors should consider that the 3 groups are included in the study :

benign breast tumour , malignant breast tumor, and control

the introduction need revision and rewritten and long paragraph should be rephrased and referenced (not enough long paragraph with one reference ) for example reference 5 and 8

ethical approval number by IRB committee should be provided

table 1 is very long you can divide it into two tables one for demographic and characteristics of studied groups and one for pathological data of malignant cases

statistical analysis section should include all statistical tests used in the study

discussion should be revised and long paragraph should be divided into more than one reference

Reviewer #3: -Inclusion criteria for patients included diagnosis with BC either malignant or benign.

All BC is malignant, please specify whether the included cases are all neoplastic or other non-neoplastic lesions were also enrolled.

-The authors aimed to evaluate the prognostic significance, however, follow up of patient to evaluate the relation with survival should have been performed.

-Have all cases been diagnosed by surgical excision of the lesion? Or other types of biopsies were performed?

-Defining ER and PR positivity is repeated, please remove one of them.

-Why were DM and Family history of cancer not reported for benign and control groups?

-Large number of pathologic data are missing. The involved pathologist could provide these data by re-evaluating HE and IHC stained slides.

-Patients’ characteristic should be presented as a separate table for control and benign groups and another one for the malignant group.

-Tables need to be thoroughly revised. Some numbers are repeated, some numbers are shifted so I was not able to assess the result of the study.

-T test is used to compare mean of two groups only. So it couldn’t be applied for comparing 3 groups as in this study. -Recheck the test and repeat the analysis.

-Correlation test used is not mentioned in statistical analysis.

-Please provide the scatter plot that demonstrate these correlations.

6. PLOS authors have the option to publish the peer review history of their article (what does this mean?). If published, this will include your full peer review and any attached files.

Reviewer #1: No

Reviewer #2: **Yes: **Rehab Ahmed Karam

Reviewer #3: **Yes: **Dina M. El-Guindy

---

## [Author Response · Author response to Decision Letter 0]

10 Mar 2023

Dear Dr.,

Thanks very much for your time. On behalf of the co-authors, I would also like to acknowledge the reviewers for their time, expertise and valuable comments that will benefit us both in the current work as well as in the future work. I have edited the manuscript to report their concerns. We wish that the manuscript is now suitable for publication in PLOS One. I included the required items when submitting my revised manuscript ('Response to Reviewers', 'Revised Manuscript with Track Changes' and 'Manuscript'.

Response to points raised by the academic editor:

My response: I am not willing to deposit our laboratory protocols in protocols.io as all the protocols used in the laboratory work of this manuscript are the manufacturer's protocols with minimal or no modifications.

My response: I checked the PLOS ONE's style requirements, including those for file naming and formatting title, authors, affiliations and edited my manuscript to meet it.

My response: I provided additional details regarding participant consent in the ethics statement (I specified that each subject was required to complete a written informed consent form) in the Methods and online submission information. 

I am not reporting a retrospective study of medical records or archived samples. So there is no need for data anonymization or patients provide informed written consent to have data from their medical records used in research.

3. We noted in your submission details that a portion of your manuscript may have been presented or published elsewhere. Please clarify whether this conference proceeding or publication was peer-reviewed and formally published. If this work was previously peer-reviewed and published, in the cover letter please provide the reason that this work does not constitute dual publication and should be included in the current manuscript.

My response: This manuscript wasn’t previously published elsewhere. An earlier draft was only posted as a preprint to research Square: doi: https://doi.org/10.21203/rs.3.rs-1925483/v1". This work does not constitute dual publication because it was posted as a preprint to allow rapid dissemination of the research findings and should be included in the current manuscript.

4. Please ensure that you include a title page within your main document. You should list all authors and all affiliations as per our author instructions and clearly indicate the corresponding author.

My response: I included a title page within my main document. I also, listed all authors and all affiliations as per PLOS One author instructions and clearly indicated the corresponding author.

5. Please include your tables as part of your main manuscript and remove the individual files. Please note that supplementary tables (should remain/ be uploaded) as separate "supporting information" files.

My response: I included the tables as part of my main manuscript and removed the individual files.

6. While revising your submission, please upload your figure files to the Preflight Analysis and Conversion Engine (PACE) digital diagnostic tool, https://pacev2.apexcovantage.com/. PACE helps ensure that figures meet PLOS requirements.

My response: I uploaded my figure files to the Preflight Analysis and Conversion Engine (PACE) digital diagnostic tool and used this tool for my figures to meet the PLOS requirements.

Response to points raised by the reviewers:

Reviewer #1: In this study, the authors evaluated expression levels of DDX43 in breast cancer cases. There some potential points that the authors should consider that;

1) The sample size is too small to conclude this conclusions!

My response: The limitation of the sample size was mentioned in the last paragraph in discussion. However, I added a new phrase to the last paragraph in discussion (To validate our findings, large-scale cohort studies are necessary) and another phrase was added to the conclusions (that DDX43 is probably under expressed in the blood of breast cancer patients in comparison to control subjects).

2) The is no correlation among gene or protein expression of DDX43 and clinical data!

My response: The correlations among gene or protein expression of DDX43 and clinical data were shown in Table 3.

3) Figure 3 is irrelevant.

My response: It shows most of the reported mRNA expression of DDX43 in various types of normal and cancer tissues including blood and breast cancer. It was added to show that DDX43 expression level is detected in normal blood & normal breast but not detected in breast adenocarcinoma and invasive lobular carcinoma. This is consistent with the present results that reported higher DDX43 mRNA expression in the blood of normal subjects than in BC patients. However, I removed it from the manuscript.

4) The authors should clarify the rationale behind the selection of DDX43 gene.

My response: We added a part in the last paragraph of introduction to clarify our rationale. (As well blood sampling is far less invasive than performing tissue biopsies for prognosis or predicting chemotherapy response. The current study has the following goals: (a) assess the DDX43 protein levels and mRNA expression in the blood of BC patients compared to control subjects) 

HAGE was shown to be expressed in various haematological and solid tumour samples but not in any of the normal tissues examined, except testis (Martelange et al, 2000; Adams et al, 2002; Mathieu et al, 2010). It had been previously reported that HAGE protein is essential for the proliferation of cancer cells and that it is immunogenic (Adams et al, 2002; Mathieu et al, 2007) [1-4]. Regarding breast cancer there are some papers had revealed that DDX43 is a potential predictor for poor prognosis, potential prognostic marker and a predictor of response to anthracycline-combination therapies (ACT) in patients with BC, and its expression is a possible prognostic marker and a predictor of response to anthracycline treatment in TNBC. [5-7]

But no previous research has investigated DDX43 blood levels of either mRNA or protein, however this is a much less invasive method for prognosis or prediction of chemotherapy response. So we aimed to investigate its mRNA and protein levels in the blood of BC patients compared to control subjects.

1. Martelange V, De Smet C, De Plaen E, Lurquin C, Boon T. Identification on a human sarcoma of two new genes with tumor-specific expression. Cancer Res. 2000;60 (14):3848–3855.

2. Adams SP, Sahota SS, Mijovic A, Czepulkowski B, Padua RA, Mufti GJ, Guinn BA. Frequent expression of HAGE in presentation chronic myeloid leukaemias. Leukemia. 2002;16 (11):2238–2242.

3. Mathieu MG, Linley AJ, Reeder SP, Badoual C, Tartour E, Rees RC, McArdle SE. HAGE, a cancer/testis antigen expressed at the protein level in a variety of cancers. Cancer Immun. 2010;10:2.

4. Mathieu MG, Knights AJ, Pawelec G, Riley CL, Wernet D, Lemonnier FA, Straten PT, Mueller L, Rees RC, McArdle SE. HAGE, a cancer/testis antigen with potential for melanoma immunotherapy: identification of several MHC class I/II HAGE-derived immunogenic peptides. Cancer Immunol Immunother. 2007;56 (12):1885–1895.

5. Chan, S. et al. “Hage (DDX43) protein expression as an independent biomarker of poor clinical outcome of breast cancer (BC) and potential as a therapeutic target for ER-negative BC.,” Journal of Clinical Oncology 2012, 30(15_suppl), pp. 1013–1013. Available at: https://doi.org/10.1200/jco.2012.30.15_suppl.1013. 

6. Abdel-Fatah TM, McArdle SE, Johnson C, Moseley PM, Ball GR, Pockley AG, Ellis IO, Rees RC, Chan SY. HAGE (DDX43) is a biomarker for poor prognosis and a predictor of chemotherapy response in breast cancer. Br J Cancer. 2014 May 13;110(10):2450-61. doi: 10.1038/bjc.2014.168.

7. Abdel-Fatah TM, McArdle SE, Agarwal D, Moseley PM, Green AR, Ball GR, Pockley AG, Ellis IO, Rees RC, Chan SY. HAGE in Triple-Negative Breast Cancer Is a Novel Prognostic, Predictive, and Actionable Biomarker: A Transcriptomic and Protein Expression Analysis. Clin Cancer Res. 2016 Feb 15;22(4):905-14. doi: 10.1158/1078-0432.CCR-15-0610.

Reviewer #2: 

- the authors should consider that the 3 groups are included in the study: benign breast tumour , malignant breast tumor, and control

My response: the benign and malignant breast cancer were added to the aim of the study, and patients and methods, 

- the introduction need revision and rewritten and long paragraph should be rephrased and referenced (not enough long paragraph with one reference) for example reference 5 and 8

My response: We had rewritten the introduction after adding updated references and the long paragraph was divided to smaller paragraphs with more references. Also, most of paragraphs from references 5 and 8 was replaced with similar data from other references.

- ethical approval number by IRB committee should be provided

My response: The ethical approval number of scientific Research Ethical Committee of Faculty of Pharmacy (Girls), Al-Azhar University was added under Ethics Statement.

- table 1 is very long you can divide it into two tables one for demographic and characteristics of studied groups and one for pathological data of malignant cases

My response: Table 1 was divided into three tables one for demographic and characteristics of studied groups and two for pathological data of benign and malignant cases.

- statistical analysis section should include all statistical tests used in the study

My response: all statistical tests used in the study were added to statistical analysis.

- discussion should be revised and long paragraph should be divided into more than one reference

My response: discussion was revised and long paragraph was divided into more than one reference

Reviewer #3: 

- Inclusion criteria for patients included diagnosis with BC either malignant or benign.

All BC is malignant, please specify whether the included cases are all neoplastic or other non-neoplastic lesions were also enrolled.

My response: We specified that both neoplastic and non-neoplastic lesions/cases were enrolled under the title (Study Population and specimen Collection). 

-The authors aimed to evaluate the prognostic significance, however, follow up of patient to evaluate the relation with survival should have been performed.

My response: the prognostic significance is removed from the abstract and aim of the study (in the last paragraph page 2) as follow up of patients to evaluate the relation with survival had not been performed due to the difficulty of following up with most patients for extended periods of time.

- Have all cases been diagnosed by surgical excision of the lesion? Or other types of biopsies were performed?

My response: all cases been diagnosed by True cut biopsy samples which is added under the title (Study Population and specimen Collection) in paragraph 4.

-Defining ER and PR positivity is repeated, please remove one of them.

My response: The repetition is removed. 

-Why were DM and Family history of cancer not reported for benign and control groups?

My response: The pathologist who collaborated in the study added these missing data.

-Large number of pathologic data are missing. The involved pathologist could provide these data by re-evaluating HE and IHC stained slides.

My response: The pathologist who collaborated in the study added most of the missing data. As a result, comparison of the mean normalized levels of serum DDX43 protein and DDX43 mRNA expression between different tumor sizes and lymph node stage was repeated.

-Patients’ characteristic should be presented as a separate table for control and benign groups and another one for the malignant group.

My response: Table 1 was divided into three tables one for demographic and characteristics of studied groups and two for pathological data of benign and malignant cases.

-Tables need to be thoroughly revised. Some numbers are repeated, some numbers are shifted so I was not able to assess the result of the study.

My response: Tables were thoroughly revised.

-T test is used to compare mean of two groups only. So it couldn’t be applied for comparing 3 groups as in this study. -Recheck the test and repeat the analysis.

My response: Surely, this is true. I checked the test and repeated the analysis using One Way Analysis of variance (ANOVA) followed by post hoc test LSD and added this part under the title (Statistical analysis).

- Correlation test used is not mentioned in statistical analysis.

My response: We added the Correlation test used under the title (Statistical analysis).

- Please provide the scatter plot that demonstrate these correlations.

My response: Scatter plot that demonstrate these correlations was added to results.

Noha Amer

On behalf of all authors.

---

## [Decision Letter · Decision Letter 1]

3 Apr 2023

DDX43 mRNA expression and protein levels in relation to clinicopathological profile of breast cancer

PONE-D-22-26586R1

Dear Dr. Noha Amer,

We’re pleased to inform you that your manuscript has been judged scientifically suitable for publication and will be formally accepted for publication once it meets all outstanding technical requirements.

Kind regards,

Elingarami Sauli, PhD

Academic Editor

PLOS ONE

Additional Editor Comments (optional):

Reviewers' comments:

Reviewer's Responses to Questions

**Comments to the Author**

1. If the authors have adequately addressed your comments raised in a previous round of review and you feel that this manuscript is now acceptable for publication, you may indicate that here to bypass the “Comments to the Author” section, enter your conflict of interest statement in the “Confidential to Editor” section, and submit your "Accept" recommendation.

Reviewer #3: All comments have been addressed

2. Is the manuscript technically sound, and do the data support the conclusions?

Reviewer #3: Yes

3. Has the statistical analysis been performed appropriately and rigorously? 

Reviewer #3: Yes

4. Have the authors made all data underlying the findings in their manuscript fully available?

Reviewer #3: Yes

5. Is the manuscript presented in an intelligible fashion and written in standard English?

Reviewer #3: Yes

6. Review Comments to the Author

Reviewer #3: Thanks for considering the reviewers’ comments. I think the manuscript has been modified and can be accepted in the current version.

7. PLOS authors have the option to publish the peer review history of their article (what does this mean?). If published, this will include your full peer review and any attached files.

Reviewer #3: **Yes: **Dina Elguindy

---

## [Editor Report · Acceptance letter]

10 May 2023

PONE-D-22-26586R1 

DDX43 mRNA expression and protein levels in relation to clinicopathological profile of breast cancer 

Dear Dr. Amer:

I'm pleased to inform you that your manuscript has been deemed suitable for publication in PLOS ONE. Congratulations! Your manuscript is now with our production department. 

Kind regards, 

on behalf of

Dr. Elingarami Sauli 

Academic Editor

PLOS ONE